# Ethylenediurea (EDU) inhibits *OsORAP1* expression in rice (*Oryza sativa* L:): Varietal differences in ozone protection efficacy

Rigyan Gupta[1,2], Shamsul H. Prodhan[1], Md. Nazmul Hasan[1], Shamsun Nahar Begum[2], Mohammad Hasanuzzaman Rani[2], Mirza Mofazzal Islam[2], Md Ashrafuzzaman[1]*

1 Department of Genetic Engineering & Biotechnology (GEB), School of Life Sciences, Shahjalal University of Science and Technology (SUST), Sylhet, Bangladesh, 2 Plant Breeding Division, Bangladesh Institute of Nuclear Agriculture (BINA), Mymensingh, Bangladesh.

* azamanbt@gmail.com or azamangeb-gen@sust.edu

## Abstract

Rising tropospheric ozone concentrations pose a significant threat to crop production in many Asian countries like Bangladesh, necessitating the development of ozone-tolerant rice (*Oryza sativa* L.) varieties. This study investigates the response of the *OsORAP1* allele—an important regulator of the plant's response to ozone stress, whose expression is associated with ozone-induced damage—in different rice varieties, particularly Kasalath-type and Nipponbare-type, under treatment with ethylenediurea (EDU), a protective antiozonant used in plant research under ambient ozone stress. The experiment, conducted during the 2022 irrigated season, involved growing of 20 rice varieties under a split-plot design with and without 300 ppm EDU treatment, followed by genomic DNA collection at the vegetative stage to differentiate Kasalath-types and Nipponbare-types, and RNA extraction from 10 selected varieties at the flowering stage for gene expression analysis. The research revealed distinct genetic responses between the two types, with Nipponbare-type varieties showing significant increases in *OsORAP1* expression and more leaf bronzing score (LBS) without EDU protection. In contrast, Kasalath-type varieties exhibited less pronounced responses due to their inherent tolerance to ozone stress. The effects of EDU on with Kasalath-type varieties responding notably in parameters such as ineffective tillers and straw yield, while Nipponbare-types showed significant changes in unfilled grains and grain yield. Additionally, differences were observed in how EDU treatment influenced reflectance indices parameters, indicating variability in how these varieties mitigate ozone stress. The study underscores the complex interaction between genetic factors, environmental conditions, and treatment in shaping the response of rice varieties to EDU, and suggests future research to further explore the genetic mechanisms, particularly the role of *OsORAP1*, that contribute to these differential responses.

**Data availability statement:** All data are in the manuscript and/or supporting information files.

**Funding:** The author(s) received no specific funding for this work.

**Competing interests:** The authors have declared that no competing interests exist.

## 1. Introduction

The presence of tropospheric ozone ($O_3$) in the atmosphere is a major threat to crop production globally. Its high phytotoxicity and occurrence above agrarian regions make it a crucial atmospheric contaminant [1–3]. The formation of $O_3$ is a result of the photochemical reactions of precursor gases that are mainly emitted from anthropogenic sources such as nitrogen oxides, carbon monoxide, volatile organic compounds, and methane [4–6]. The adverse effects of ambient $O_3$ can lead to estimated yield losses of 4% globally and 10% regionally for rice [2,7] and 4–17% for wheat, 10–14% for soybean, and 3–6% for maize, resulting in a projected annual economic loss of US$ 12–21 billion based on global crop models [8].

The harmful effects of $O_3$ on vegetation were first identified in North America during the 1950s and later observed in Europe and Japan in the 1970s [5,9]. Ground-level $O_3$ concentrations have significantly increased in urban areas and are even higher in rural areas than in cities [10,11]. The first recorded incident of photochemical smog and $O_3$ damage to vegetation was reported by Middleton *et al.* [12], while Richards *et al.* [13] found $O_3$ as a phytotoxic constituent of photochemical smog while investigating leaf injury in grape vines (*Vitis vinifera*) in Southern California.

Upon entering plant stomata, $O_3$ dissolves in the apoplast and generates different reactive oxygen species (ROS). These ROS can change cellular functions that result in premature senescence, cell death, and the up- or down-regulation of specific genes [14,15]. Rice (*Oryza sativa* L.) is a staple crop in Asia but is severely affected by tropospheric $O_3$, particularly in South and East Asia. Since its cropping season overlaps with peak $O_3$ concentrations, breeding $O_3$-tolerant rice varieties is necessary to ensure global food supply [16] as the demand for food is projected to double by 2050. The chemical compound Ethylenediurea (EDU; N- [2-(2-oxo-1- imidazolidinyl) ethyl]-N′-phenylurea) was initially introduced by Carnahan, Jenner, and Wat in 1978 [17] and has been utilized for a considerable time to assess the effects of $O_3$ on plants [18–20]. EDU is conveniently sprayed on leaves, works as an antiozonant, and provides a "control" without $O_3$ damage. The mode of action of EDU against $O_3$ is still unclear. It can be used to screen plants for $O_3$ sensitivity under ambient conditions, verify foliar $O_3$ symptoms in the field, and thus suppress $O_3$ injury in plants [19,21].

Previous research has identified genetic factors associated with $O_3$ tolerance in rice, suggesting that this trait is controlled by multiple medium-effect loci rather than a single large-effect locus [7,22]. In a study by Frei *et al.* [23] using a mapping population derived from the cultivars Nipponbare (susceptible) and Kasalath (tolerant), certain $O_3$-related quantitative trait loci (QTLs) were identified. One of the identified QTLs, OzT9, was associated with visible leaf symptom formation in response to $O_3$ stress, where the allele from the tolerant cultivar Kasalath conferred tolerance. The effectiveness of OzT9 was supported by SL41, a chromosomal segment substitution line carrying a chromosomal introgression from Kasalath at the OzT9 locus in the Nipponbare background, which showed fewer visible symptoms than Nipponbare [23]. Further transcriptomic analysis revealed differential gene expression profiles between Nipponbare and SL41A under $O_3$ stress [24]. Physiologically, apoplastic

ascorbate (ASA) is a line of defense against $O_3$ entering the plant through stomata. Several studies, including Luwe *et al.* [25], Plöchl *et al.* [26], and Feng *et al.* [27], support this connection between the annotation of candidate genes and $O_3$ stress tolerance. Moreover, a previous study showed that an AO-overexpressing tobacco plant affects $O_3$ tolerance adversely due to altered apoplastic reduced ASA content and ASA redox conditions [28]. This transformative evidence suggests that the expression of the putative AO gene is also associated with $O_3$ stress tolerance in rice and the formation of significant symptoms.

According to a study by Frei *et al.* [24], the expression of the *OsORAP1* (*OZONE RESPONSIVE APOPLASTIC PROTEIN1*) gene in response to $O_3$ stress was significantly lower in the $O_3$-tolerant SL41 variety compared to the cv. Nipponbare. The gene was located near a QTL and showed differential regulation between a sensitive parent and a tolerant introgression line. Additional analysis revealed that the promoter region of *OsORAP1* had widespread polymorphisms, with the OzT9 donor variety responsible for this variation between the $O_3$-susceptible cv. Nipponbare and the $O_3$-tolerant cv. Kasalath. *OsORAP1* encodes an apoplastic protein that is involved in cell death processes but has no ascorbate oxidase activity. The lower or no expression of *OsORAP1* enhanced $O_3$ tolerance in rice, leading to its designation as *OZONE RESPONSIVE APOPLASTIC PROTEIN1*. A previous study utilized a transgenic T-DNA insertion line to suppress *OsORAP1* expression and confer $O_3$ tolerance. However, it is unclear whether this suppression can also be achieved naturally through polymorphism. Generally, the Nipponbare type allele, carried by a group of rice varieties, showed a significantly higher level of foliar symptom formation, lipid peroxidation, and biomass loss when exposed to elevated $O_3$ in a controlled setting [24]. Furthermore, the Nipponbare types exhibited significantly higher induction of *OsORAP1* expression in response to $O_3$ stress, suggesting that *OsORAP1* plays a crucial role in regulating $O_3$ response and susceptibility or tolerance in rice.

Previous studies have explored the use of ethylenediurea (EDU) as a protective agent against ozone stress in rice. Several researchers have reported that EDU can enhance rice yield, primarily through improvements in grain weight rather than increases in the number of panicles or grains per plant [29]. Under conditions of elevated surface ozone, EDU has been shown to provide moderate mitigation of yield loss, suggesting its potential as a practical tool for safeguarding rice productivity [30,31]. Interestingly, its effectiveness appears to vary between rice genotypes, with some evidence indicating that inbred cultivars benefit more from EDU treatment than hybrid varieties [32]. In addition to its physiological effects, EDU application does not appear to significantly alter gene expression patterns, highlighting its suitability for field-based ozone screening without interfering with molecular responses. More recently, Frei et al. [16] demonstrated that EDU can serve as a diagnostic tool to differentiate between ozone-sensitive and ozone-tolerant rice genotypes, with yield increases observed up to 21% in responsive varieties. These findings collectively underscore the potential of EDU in both protecting rice from ozone stress and facilitating the identification of tolerant cultivars. These studies provide a strong background for our work and highlight the rationale for using EDU in our experiment.

In addition, we have now clearly stated the limitation of the present study. While the study identifies ozone-tolerant rice varieties and explores their genetic expression profiles, it is limited to a single growing season and location. Furthermore, the effect of EDU was studied at only one concentration (300 ppm), and the long-term agronomic and physiological impacts under varying field conditions were not assessed. These aspects should be explored in future multi-location and multi-season trials. The study was conducted based on 10 Bangladeshi rice varieties during their flowering stage with and without the application of EDU (300 ppm, once a week after transplanting until harvesting). Therefore, the aim of this study was to investigate the impact of $O_3$ stress on rice and to assess the potential of EDU as a bio-monitoring tool for $O_3$ in highly affected developing countries such as Bangladesh. Since conventional $O_3$ experimental facilities like open top chamber (OTC) and free-air concentration enrichment (FACE) $O_3$ are lacking in these countries, this study aimed to evaluate the expression of *OsORAP1* gene and its correlation with $O_3$ tolerance when the plants are continuously exposed to an average of 70 ppb or more ambient $O_3$ applying EDU by collecting the leaf samples during the flowering stage to assess gene expression.

   

## 2. Materials and methods

### 2.1 Experimental site and plant materials

The experiment was set in Bangladesh Institute of Nuclear Agriculture (BINA) HQs, Mymensingh farm (24° 43' N latitude and 90° 25' E longitude), Bangladesh which is situated in Old Brahmaputra Floodplain (AEZ-23) at irrigated (boro) season of 2022 (which is denoted as dry season usually depend on irrigated water and the crops transplanted from December to early February and harvested between April and June). A total of twenty high yielding popular (inbred and hybrid) and local diversified *indica and Aus* rice genotypes were used in this experiment, and 10 (Binadhan-15, Binadhan-17, BR11, Hutra, Kasalath, Binadhan-12, Binadhan-16, BRRI dhan28, BRRI dhan29, and BRRI hybrid dhan3) representative were selected for gene expression analysis with details provided in Table 1.

### 2.2 Growing of varieties, design of the experiments and measurement of data

The seeds of twenty varieties were wrapped up in water container for 24 hours. After that, the seeds were removed from water and kept in gunny bags and leave in dark room. It was required about 3–5 days to germinate the all seed samples. Then, the germinated seed were sown in nursery beds with definite spacing and well tagged with a good layout. The experimental fields were prepared by ploughing and laddering with a tractor. By removing all the stubbles and weeds the field was then prepared. Then thirty-five day old seedlings were detached from the nursery bed with healthy root system. The experiments were than laid out as split-plot designs with four replicates arranged in blocks, including EDU treatment as the main plot (i.e., with and without EDU), and genotype as the subplot. Each subplot was 4 m² (2 × 2 m) size of which a central one m² area were used for yield estimation at harvest and the border plants may be used for destructive

**Table 1. Brief description, collection and/or origin of the rice varieties investigated in this experiment carrying different alleles at the *OsORAP1* locus using primer KAS_1_2 [33].**

| Sl. No. | Variety name | Short description | Collection and/or origin | Subspecies | *OsORAP1* allele |
|---|---|---|---|---|---|
| 1 | Iratom-24 | Early maturing high-yielding variety (HYV) | BINA/ Bangladesh | *Indica* | Nipponbare-type |
| 2 | Binadhan-6 | HYV | BINA/ Bangladesh | *Indica* | Kasalath-type |
| 3 | Binadhan-10 | Salt tolerant HYV | BINA/ Bangladesh | *Indica* | Nipponbare-type |
| 4 | Binadhan-11 | Submergence tolerant | BINA/ Bangladesh | *Indica* | Nipponbare-type |
| 5 | Binadhan-12 | Submergence tolerant (fine grain) | BINA/ Bangladesh | *Indica* | Nipponbare-type |
| 6 | Binadhan-14 | Late sowing potential HYV | BINA/ Bangladesh | *Indica* | Kasalath-type |
| 7 | Binadhan-15 | Premium quality | BINA/ Bangladesh | *Indica* | Kasalath-type |
| 8 | Binadhan-16 | Short duration HYV | BINA/ Bangladesh | *Indica* | Nipponbare-type |
| 9 | Binadhan-17 | HYV | BINA/ Bangladesh | *Indica* | Kasalath-type |
| 10 | BR11 | Widely grown HYV | BRRI/ Bangladesh | *Indica* | Kasalath-type |
| 11 | BRRI dhan28 | Early maturing HYV | BRRI/ Bangladesh | *Indica* | Nipponbare-type |
| 12 | BRRI dhan29 | HYV | BRRI/ Bangladesh | *Indica* | Nipponbare-type |
| 13 | BRRI dhan88 | Early maturing HYV | BRRI/ Bangladesh | *Indica* | Nipponbare-type |
| 14 | BRRI hybrid dhan1 | High-yielding hybrid variety | BRRI/ Bangladesh | *Indica* | Nipponbare-type |
| 15 | Hutra | Landrace | Chapainawabganj/ Bangladesh | *Aus* | Kasalath-type |
| 16 | Tepiboro | Landrace | Chapainawabganj/ Bangladesh | *Aus* | Kasalath-type |
| 17 | BRRI hybrid dhan3 | High-yielding hybrid variety | BRRI/ Bangladesh | *Indica* | Nipponbare-type |
| 18 | IR64 | International mega-variety | BRRI/ IRRI, Philippines | *Indica* | Nipponbare-type |
| 19 | Kasalath | Landrace | BRRI/ Bangladesh | *Aus* | Kasalath-type |
| 20 | BRRI dhan58 | HYV | BRRI/ Bangladesh | *Indica* | Nipponbare-type |

Note: BINA = Bangladesh Institute of Nuclear Agriculture, BRRI = Bangladesh Rice Research Institute and IRRI = International Rice Research Institute

sampling during the growing season. After that, the seedling were transplanted in the main field at the rate one seedling per hill with a spacing 20 cm row to row distance and 15 cm hill to hill distance. Considering two treatments and four replicates, each experimental field was approximate size of 1050 m² (1000 m² experimental area plus dikes, bunds, irrigation channels, etc.). Urea, triple super phosphate, muriate of potash, gypsum, zinc sulphate and boron (the main source of N, P, K, S, Zn and B respectively) with a dose @ 270 Kg, 115 Kg, 150 Kg, 75 Kg, 6 Kg and 7 Kg, respectively were applied in the experimental field. During the final land preparation, Carbofuran-5G @ 10 Kg/ha (a granular insecticide) and all fertilizers excluding urea and zinc sulphate were applied; one-third of the urea dose was applied before the full dose of zinc sulphate was applied 12 days after transplanting. Again, urea were used in two split doses at 30 and 45 days after transplanting. For the reductions of insect and other pest infestation the pesticides were applied as when necessary. Three times of weeding were done at 15, 30 and 45 DAT.

Before each measurement, five randomly selected plants from each subplot were tagged. After harvesting, data were collected from a 1 m² area in each subplot, along with the five selected plants. The measured parameters included plant height (cm), total number of tillers per plant, number of effective and non-effective tillers per plant, panicle length (cm), filled grain count per plant, filled grain weight per plant, unfilled grain count per plant, unfilled grain weight per plant, grain yield, straw yield, and thousand-kernel weight (S2 dataset). The harvest index (%) was measured by using the following formula:

i)  Harvest index (%) = (Grain yield)/(Total biomass) × 100

A leaf bronzing score (LBS) ranging from 0 to 10 was used to assess ozone-induced leaf damage, following the method described by Ueda et al. [7]. A score of 0 represented no visible symptoms on any leaves, while a score of 10 indicated severe damage affecting the entire plant due to ozone stress.

Stomatal conductance was assessed using a leaf porometer (Model SC1, Decagon Devices, Pullman, WA) on the youngest fully expanded leaves of a selected plant per subplot. Measurements were taken on sunny days between 9:30 AM and 2:00 PM. To evaluate ozone stress, spectral reflectance was recorded (S3 dataset) with a handheld spectroradiometer (Polypen RP410, PRI, Czech Republic). The normalized difference vegetation index (NDVI) was calculated using the formula:

ii)  NDVI = $(R_{780} - R_{630})/ (R_{780} + R_{630})$.

Similarly, the Lichtenthaler Index 2 (Lic2) was determined as the ratio of $R_{440}$ to $R_{690}$, i.e.,

iii) Lic2 = $R_{440} / R_{690}$

iv) Simple ratio index (SR) = $R_{NIR}/R_{RED}$

v) Photochemical reflectance index (PRI) = $(R_{531} - R_{570})/(R_{531} + R_{570})$

vi) Anthocyanin reflectance index (ARI) = $(R_{550})^{-1} - (R_{700})^{-1}$.

Here, in the vegetation indices, R refers to reflectance and subscript indicates the wave bands in nanometers.

## 2.3  Plant material collection for extraction of Genomic DNA and analysis

Young, vigorously growing fresh leaf samples from the 20 rice genotypes were collected from the seedlings at vegetative stage (85 days old seedlings) to extract genomic DNA. Initially, healthy portion of the youngest leaves of the seedlings were cut apart with sterilized scissors and washed in distilled water and ethanol and dried on fresh tissue paper to remove spore of microorganisms and any other source of foreign DNA. The collected leaf samples were then put into polythene bags and kept on ice in an icebox. Then, the polythene bags were wrapped in aluminium foil and stored at −80⁰C freezer.

Genomic DNA of the selected rice genotypes were extracted from leaf samples using a DNeasy Plant Mini Kit (Qiagen, Düsseldorf, Germany). For confirmation of genomic DNA, agarose gel electrophoresis were done by visualizing the

DNA band in the Gel Doc system (Biometra, Japan). For the determination of quantity and purity the extracted samples were analyzed with Nanodrop 2000c (Thermo Fisher Scientific Inc., Wilmington, DE, USA). Then the DNA concentrations were adjusted to 25 ng/ μl. Then, PCR reactions were performed on each DNA sample in a 10μl reaction mix containing the following reagents: 1μl Ampli Taq polymerase buffer (10X), 2.5μl Primer (10μM), 1μl dNTPs (250μM), 0.2μl Ampli Taq DNA polymerase, 2μl Genomic DNA (25ng/μl) and 3.3μl Sterile deionized water. DNA amplification were performed in an oil-free thermal cycler. The reaction mix were preheated at $94^0$C for 5 minutes followed by 35 cycles of 1 min denaturation at $94^0$C, 1 min annealing at $54^0$C and, elongation or extension at $72^0$C for 2 minutes. After the last cycle, a final step for 7 minutes at $72^0$C were allowed to complete the extension of all amplified fragments. After completion of the cycling program, reactions were held at $4^0$C. The primer KAS_1_2 sequence that were used for the initial screening is 5′-GCCTTCCTCCTTGTGGTCG-3′ (forward) and 5′- AGGGAACGTCCCACTGTTG-3′ (reverse) for the Kasalath-type *OsORAP1* allele (annealing temperature 57 °C).

Again, agarose gel electrophoresis were done by ethidium bromide staining for visualizing the band (Kasalath-type) in the Gel Doc system (Biometra, Japan). Among the twenty genotypes the selection was based on the presence and absence of a PCR band [33] for further analyses (Table 1 and S1 Fig). Based on PCR bands we classified the varieties into two distinct clades: Kasalath-types and Nipponbare-types and found that among the 20 rice varieties, 8 varieties showed PCR bands of were Kasalath-type *OsORAP1* allele and the remaining 12 varieties which were not show PCR bands of were Nipponbare-type *OsORAP1* allele [33].

## 2.4 Ozone treatment and monitoring

After transplanting, the two treatments viz. with EDU (Ethylenediurea) and without EDU (control) were applied and plants were continuously exposed to natural field conditions of BINA HQs farm, Mymensingh, Bangladesh where the ambient ozone concentration was more than 70 ppb, quantified by a well handheld ozone device (series 500; Aeroqual Ltd. Auckland, New Zealand) at every day after transplanting (S1 dataset). The chemical EDU at the concentration of 300 ppm once a week were applied as a foliar spray in the morning (9 am to 12 pm) to mitigate the ozone effects [16]. The controls plant were sprayed with similar of water which was needed for spraying EDU.

## 2.5 Collection of leaf samples for RNA isolation and gene expression analyses

At the flowering stage (115 days old) between 10 a.m. and 12 p.m., the leaf samples of the selected rice varieties viz. 5 Kasalath-types varieties (Binadhan-15, Binadhan-17, BR11, Hutra, and Kasalath) and 5 Nipponbare–type (Binadhan-12, Binadhan-16, BRRI dhan28, BRRI dhan29, and BRRI hybrid dhan3) were collected. Two randomly selected plants of each genotype from each treatment plot were collected and leaves were pooled for one representative sample. Then the samples were flash-frozen in liquid nitrogen and stored at −80 °C for RNA extraction.

Again, each genotype among the 10 genotypes was extracted from 3 replicates (of 4) of each treatment, yielding a total of 6 samples. In the next step, whole leave samples were ground in liquid nitrogen to a fine powder, and total RNA was extracted with the Promega RNA Isolation Kit containing RQ1 DNase (Promega). In order to determine the quantities and purity of the extracted samples, spectrophotometers NanoDropTM 2000/2000c (Thermo Fisher Scientific Inc., Wilmington, DE, USA) was used. Reverse transcription of 300 ng of total RNA was performed using TAKARA PrimeScriptTM 1st strand cDNA Synthesis Kit (Cat. # 6110A). In accordance with the manufacturer's instructions, quantitative PCR was performed with TAKARA TB Green Premix Ex Taq (Tli RNase H Plus) (Cat. # RR420A). In the reaction mixture, there were 7.5μl of TB Green Premix Ex Taq (Tli RNaseH Plus) (2X), 0.4μl of each gene-specific primer (10 μM), 0.2μl of ROX Reference Dye (50X), 4.5μl of nuclease-free water, and 2μl of cDNA Template (15 ng) sample. The following thermal conditions prevailed: denaturation at 95 °C for 10 min, 40 cycles of 15s of denaturation at 95 °C, and 1 min of annealing and extension at 60 °C. After each reaction, melting curves were analyzed to verify primer specificity. In this study, the housekeeping gene named 18S rRNA (AK059783) was amplified using 5′-CTACGTCCCTGCCCTTTGTACA-3′

(forward) and 5´-ACACTTCACCGGACCATTCAA-3´ (reverse) cast-off as an endogenous reference [34]. A delta-delta (Δ-Δ) CT quantification method was applied to quantify target genes (S4 Dataset) according to Frei *et al.* [24]. Based on a cDNA dilution series, the efficiency of amplification was more than 90% for the primer pair. Analytical duplicates were used for gene expression analyses. An expression analysis of *OsORAP1* was conducted using the primer pair's 5´-CATCGAGGCGCACTTCTT-3´ (forward) and 5´-CGAACGGCTAGCTTCTGGT-3´ (reverse).

## 2.6 Statistical analysis

For analysis of variance the experimental data were subjected to mixed model analysis using the package lme4 in R [35] with Satterthwaite's method where EDU treatment, variety, and their interaction were set as fixed effects. Pairwise comparisons of individual varieties with and without EDU treatment were obtained using the procedure emmeans in R and adjusted using the Tukey's method. The total analysis was conducted in R 4.2.1 (R Core Team 2022) with R packages lmerTest [36], multcomp [37] and performance [38].

## 3. Results

### 3.1 Ozone concentrations levels in study site

During whole growing period the average ozone concentrations were more than 70 ppb (76.49 ppb) where the threshold level of ozone for crop production is 40 ppb [39]. Daily average ozone concentration were given in Fig 1 where it was found that the ozone concentrations was more than the threshold level which drastically affects the production of rice in Mymensingh, Bangladesh.

### 3.2 Varietal differences in the responsiveness of *OsORAP1* expression to EDU treatment in rice

This study analyzed the *OsORAP1* locus among different rice varieties and revealed two distinct clades of varieties: the Nipponbare-type and the Kasalath-type allele. To investigate the effect of EDU on mRNA expression, 10 rice varieties were grown and exposed to ambient $O_3$ stress in a field condition where the $O_3$ concentration was higher than 70 ppb. The Kasalath and Nipponbare types both showed highly significant (P<0.001) *OsORAP1* expression in EDU treatment, variety and, EDU treatment and varietal interaction which indicate the EDU have effect on *OsORAP1* expression. (Table 2).

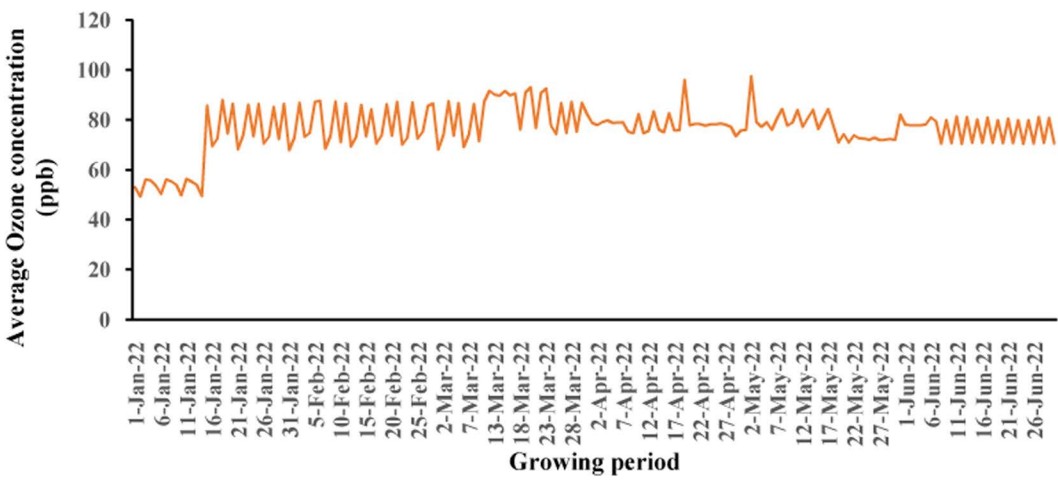

**Fig 1. Daily average ozone concentration in experimental location (Mymensingh) in the *Boro* rice growing seasons 2022.**

**Table 2. Analysis of variance (ANOVA) of different traits (Molecular, Agronomic and physiological) of ten selected rice varieties grown in ozone-polluted location of Mymensingh, Bangladesh in irrigated '22 season with the application of EDU or without application of EDU (control).**

| Traits | ANOVA | | | | | | Treatment Means | |
|---|---|---|---|---|---|---|---|---|
| | EDU-Treatment | | Variety | | Interaction | | Control | EDU |
| | Pr(>F) | F values | Pr(>F) | F values | Pr(>F) | F values | | |
| **Kasalath-type** | | | | | | | | |
| Molecular | | | | | | | | |
| *OsORAP1* Expression | <0.000*** | 19.483 | <0.000*** | 30.415 | <0.000*** | 31.076 | 0.567 | 1.508 |
| Agronomic | | | | | | | | |
| Leaf bronzing score (lbs) | 1.000 | 0.000 | <0.000*** | 6.565 | 0.074 | 2.411 | 2.400 | 2.400 |
| No. of effective tiller plant$^{-1}$ | 0.081 | 3.272 | 0.032* | 3.038 | 0.157 | 1.790 | 9.170 | 9.950 |
| No. of ineffective tiller plant$^{-1}$ | 0.023* | 5.845 | 0.002** | 5.681 | 0.824 | 0.376 | 0.800 | 0.570 |
| No. of filled grain plant$^{-1}$ | <0.000*** | 17.871 | <0.000*** | 42.807 | 0.723 | 0.518 | 1362.000 | 1522.490 |
| No. of unfilled grain plant$^{-1}$ | 0.366 | 0.847 | <0.000*** | 22.610 | 0.250 | 1.433 | 537.120 | 515.580 |
| Thousand kernel weight (TKW) | 0.783 | 0.077 | <0.000*** | 1536.219 | 0.979 | 0.107 | 24.076 | 24.063 |
| Grain yield plant$^{-1}$ | 0.144 | 2.253 | 0.001** | 5.902 | 0.020* | 3.435 | 274.333 | 291.318 |
| Straw yield plant$^{-1}$ | <0.000*** | 7.729 | <0.000*** | 8.136 | 0.404 | 1.038 | 18.824 | 20.987 |
| Harvest index (%) | 0.554 | 0.358 | 0.015* | 3.661 | 0.313 | 1.244 | 52.316 | 51.433 |
| Physiological | | | | | | | | |
| NDVI | 0.052 | 3.868 | <0.000*** | 20.637 | 0.044 | 2.539 | 0.690 | 0.698 |
| SR | 0.028* | 4.950 | <0.000*** | 20.622 | 0.053 | 2.411 | 5.503 | 5.683 |
| SIPI | 0.245 | 1.368 | <0.000*** | 24.909 | 0.111 | 1.928 | 0.754 | 0.756 |
| Lic2 | 0.010* | 6.803 | <0.000*** | 10.389 | 0.376 | 1.068 | 0.675 | 0.690 |
| PRI | 0.432 | 0.620 | <0.000*** | 8.533 | 0.250 | 1.367 | 0.055 | 0.056 |
| **Nipponbare-type** | | | | | | | | |
| Molecular | | | | | | | | |
| *OsORAP1* Expression | <0.000*** | 65.595 | <0.000*** | 15.000 | <0.000*** | 9.512 | 3.839 | 1.270 |
| Agronomic | | | | | | | | |
| Leaf bronzing score (lbs) | <0.000*** | 128.000 | <0.000*** | 21.375 | 0.363 | 1.125 | 3.900 | 2.300 |
| No. of effective tiller plant$^{-1}$ | 0.670 | 0.185 | 0.035* | 2.957 | <0.000*** | 8.999 | 10.270 | 10.090 |
| No. of ineffective tiller plant$^{-1}$ | 0.364 | 0.850 | 0.049* | 2.701 | 0.935 | 0.202 | 0.870 | 0.710 |
| No. of filled grain plant$^{-1}$ | 0.172 | 1.962 | <0.000*** | 122.829 | <0.000*** | 4.429 | 1563.730 | 1490.400 |
| No. of unfilled grain plant$^{-1}$ | 0.009** | 7.711 | <0.000*** | 19.897 | 0.177 | 1.696 | 556.510 | 385.123 |
| Thousand kernel weight (TKW) | 0.154 | 2.153 | <0.000*** | 2387.456 | 0.408 | 1.033 | 23.263 | 23.096 |
| Grain yield plant$^{-1}$ | 0.031* | 5.176 | <0.000*** | 35.249 | 0.035* | 3.026 | 245.273 | 268.394 |
| Straw yield plant$^{-1}$ | 0.018 | 6.345 | <0.000*** | 11.885 | 0.066 | 2.502 | 20.394 | 24.297 |
| Harvest index (%) | 0.507 | 0.451 | <0.000*** | 14.024 | 0.243 | 1.448 | 51.895 | 53.072 |
| Physiological | | | | | | | | |
| NDVI | 0.219 | 1.527 | <0.000*** | 6.091 | 0.192 | 1.554 | 0.672 | 0.680 |
| SR | 0.304 | 1.067 | <0.000*** | 6.920 | 0.138 | 1.780 | 5.202 | 5.318 |
| SIPI | 0.180 | 1.818 | <0.000*** | 6.987 | 0.380 | 1.060 | 0.738 | 0.744 |
| Lic2 | 0.632 | 0.231 | 0.012* | 3.328 | 0.146 | 1.743 | 0.682 | 0.686 |
| PRI | 0.267 | 1.241 | <0.000*** | 9.358 | 0.893 | 0.276 | 0.041 | 0.045 |

Note: NDVI = Normalized difference vegetation index, SR = Simple ratio, SIPI = Structure intensive pigment index, Lic2 = Lichtenthaler indices 2, PRI = Photochemical reflectance index; Asterisks indicate the level of statistical significance *, P < 0.05; **, P < 0.01; ***, P < 0.001; no asterisks means not significant.

Further analysis showed that within the varieties, Kasalath-type varieties showed less *OsORAP1* expression in both EDU treated varieties and control varieties except Kasalath, while the Nipponbare-types showed higher significant *OsORAP1* expression in control varieties than EDU treated varieties except Binadhan-16 and BRRI hybrid dhan3 which showed higher *OsORAP1*expression but not significant (Fig 2). These results suggest that the expression of *OsORAP1* is more responsive to EDU application in Nipponbare-type varieties than Kasalath-type varieties.

### 3.3 Responses to ambient O$_3$ stress of the varieties carrying different alleles at *OsORAP1* locus due to EDU application

**3.3.1. Leaf bronzing score.** Upon prolonged exposure to ambient O$_3$ stress during the flowering or late flowering stage, visible symptoms of O$_3$ damage were observed. The formation of visible symptoms due to ambient ozone stress was found to differ significantly between the two allele groups when EDU treatments were applied. Nipponbare-type varieties displayed highly significant response, while Kasalath-type varieties showed insignificant response in EDU treatment. Again, Kasalath-type varieties showed insignificant response within the varieties and interaction between varieties and EDU treatment. On the other hand, Nipponbare-type varieties showed highly significant response within the variety but insignificant between varietal and EDU treatment interactions (Table 2).

In Nipponbare-type varieties, the response of EDU was prominent and the score in control plots were higher and significant, where the EDU treated varieties plots scores were less. Kasalath-type varieties response to EDU treatment were not significant except Kasalath because this variety is highly tolerant to ozone stress [16,40] and not responsive to EDU this is why the EDU treated Kasalath variety gave significant response while the control Kasalath variety did not (Fig 3).

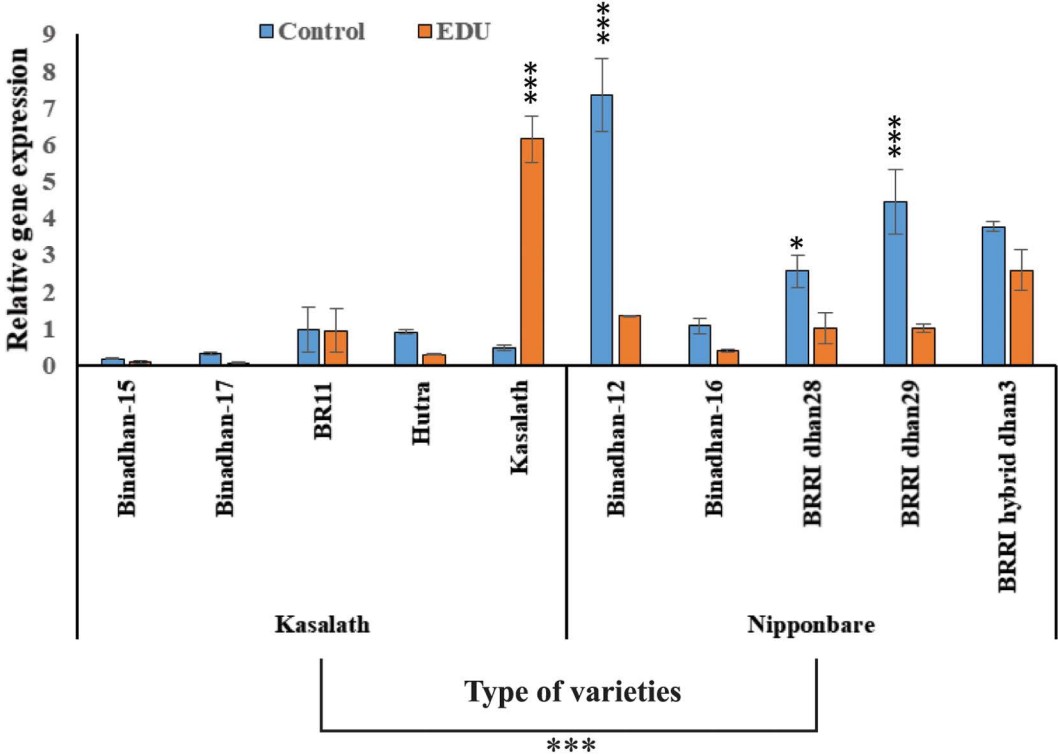

**Fig 2. Expression of *OsORAP1* in 5 Kasalath-type and 5 Nipponbare-type rice varieties exposed to ambient ozone stress with the treatment of EDU and without EDU (control) at Real-time PCR data.** Means values and standard errors (n = 3) are shown. Asterisks indicate significant differences. *, P < 0.05; **, P < 0.01; ***, P < 0.001; no asterisks means not significant.

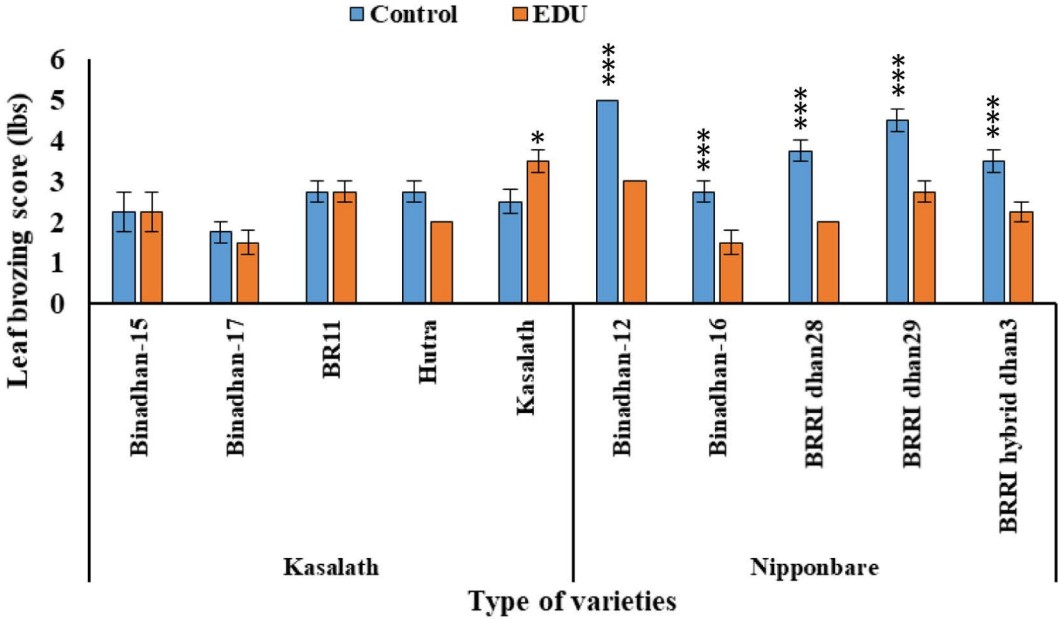

**Fig 3. Leaf bronzing scores of 5 Kasalath-type and 5 Nipponbare-type rice varieties exposed to ambient ozone concentration with and without application of ethylenediurea (EDU); bars indicate mean values of Mymensingh for Boro season 2022 with four replicates and standard errors (n = 4); asterisks above the bars indicate significant difference between control and EDU treatment with one variety by Tukey's test (* p < 0.05, **p < 0.01, ***p < 0.001).**

**3.3.2. Yield parameters.** The effects of Ethylenediurea (EDU) on yield parameters were investigated in both Kasalath-type and Nipponbare-type rice varieties, revealing varying responses across different parameters and varieties. In Kasalath-type varieties, significant responses to EDU treatment were observed for parameters including the number of ineffective tillers plant$^{-1}$, number of filled grains plant$^{-1}$, and straw yield plant$^{-1}$. However, other parameters such as the number of effective tillers plant$^{-1}$, number of unfilled grains plant$^{-1}$, thousand kernel weight (TKW), grain yield plant$^{-1}$, and harvest index (%) did not display significant responses to EDU treatment. Within the Kasalath-type varieties, responses were highly significant for all studied characters, but the interaction between EDU treatment and varieties was not significant except for grain yield plant$^{-1}$. Similarly, in Nipponbare-type varieties, significant responses to EDU treatment were observed for parameters such as the number of unfilled grains plant$^{-1}$ and grain yield plant$^{-1}$. Conversely, other parameters like the number of effective tillers plant$^{-1}$, number of ineffective tillers plant$^{-1}$, number of filled grains plant$^{-1}$, TKW, straw yield plant$^{-1}$, and harvest index (%) did not exhibit significant responses to EDU treatment. Within the Nipponbare-type varieties, responses were highly significant for all studied characters, and the interaction between EDU treatment and varieties was highly significant for parameters including the number of effective tillers plant$^{-1}$, number of filled grains plant$^{-1}$, and grain yield plant$^{-1}$ and other parameters were not significantly response (Table 2).

Specifically focusing on grain yield plant$^{-1}$ and straw yield plant$^{-1}$, Kasalath-type varieties generally did not exhibit significant gains or losses in response to EDU treatment, except for BR11, which showed a significant increase in both parameters. Conversely, in Nipponbare-type varieties, BRRI hybrid dhan3 and BRRI dhan29 exhibited significant gains in grain yield plant$^{-1}$ and straw yield plant$^{-1}$ (S2 Fig) in response to EDU treatment, while other varieties did not show significant changes in either parameter (Table 3, Fig 4a and 4b).

Regarding the parameter harvest index (%), neither Kasalath-type nor Nipponbare-type varieties exhibited significant gains or losses due to EDU treatment (Table 3, Fig 4c). So, the variability in the response of different rice varieties to EDU

**Table 3. Parameters of contrasting rice varieties exposed to control (without EDU) and EDU treated varieties in ambient ozone stress conditions.**

| Sl. No. | Name of the varieties | Grain yield plant⁻¹ (g) | | | Straw yield plant⁻¹ (g) | | | Harvest Index (%) | | |
|---|---|---|---|---|---|---|---|---|---|---|
| | | Control (Without EDU) | EDU treated | Gain/Loss (%) | Control (Without EDU) | EDU treated | Gain/Loss (%) | Control (Without EDU) | EDU treated | Gain/Loss (%) |
| **Varieties with Kasalath-type *OsORAP1* allele** | | | | | | | | | | |
| 1 | Binadhan-15 | 27.11 (±1.06) | 28.76 (±1.71) | 6.06 (±9.75) | 16.63 (±0.31) | 19.08 (±2.09) | 14.75 (±13.60) | 53.11 (±1.70) | 49.48 (±2.21) | −6.84 (±7.26) |
| 2 | Binadhan-17 | 27.11 (±0.98) | 26.74 (±0.99) | −1.37 (±6.17) | 19.47 (±0.38) | 20.29 (±1.92) | 4.22 (±11.37) | 57.95 (±2.52) | 56.90 (±2.73) | −1.80 (±1.34) |
| 3 | BR11 | 27.92 (±2.81) | 37.66 (±1.82) | 34.83 (±23.35) | 20.85 (±0.39) | 25.89 (±0.77) | 24.18 (±4.77) | 47.69 (±1.70) | 53.17 (±1.62) | 11.49 (±6.22) |
| 4 | Hutra | 24.63 (±0.98) | 24.33 (±1.73) | −1.23 (±8.37) | 16.35 (±0.93) | 18.21 (±0.65) | 11.39 (±5.59) | 51.91 (±2.11) | 49.13 (±1.86) | −5.36 (±3.51) |
| 5 | Kasalath | 30.38 (±2.56) | 28.17 (±2.07) | −7.26 (±14.51) | 20.82 (±1.90) | 21.46 (±1.20) | 3.07 (±8.06) | 50.92 (±4.05) | 48.49 (±1.70) | −4.78 (±11.13) |
| **SEm (±)** | | 1.68 | 1.66 | 12.43 | 0.78 | 1.29 | 8.68 | 2.42 | 2.03 | 5.89 |
| **Mean** | | 27.43 | 29.13 | 6.20 | 18.82 | 20.99 | 11.51 | 52.32 | 51.43 | −1.91 |
| **Varieties with Nipponbare-type *OsORAP1* allele** | | | | | | | | | | |
| 6 | Binadhan-12 | 23.82 (±1.64) | 23.26 (±1.76) | −1.79 (±12.70) | 19.19 (±1.41) | 24.55 (±1.56) | 27.93 (±5.81) | 51.34 (±2.26) | 47.72 (±2.45) | −7.05 (±7.91) |
| 7 | Binadhan-16 | 25.40 (±1.06) | 26.09 (±0.46) | 2.71 (±5.67) | 15.19 (±1.33) | 11.66 (±0.78) | −23.25 (±7.48) | 59.62 (±1.23) | 64.08 (±1.08) | 7.48 (±2.56) |
| 8 | BRRI dhan28 | 18.16 (±1.64) | 16.48 (±0.65) | −9.26 (±6.25) | 21.15 (±2.54) | 21.23 (±2.67) | 0.38 (±19.28) | 44.69 (±3.93) | 41.73 (±4.07) | −6.63 (±14.02) |
| 9 | BRRI dhan29 | 23.05 (±2.53) | 28.65 (±2.81) | 24.33 (±19.69) | 25.01 (±1.96) | 33.66 (±6.24) | 34.61 (±39.28) | 46.46 (±4.03) | 53.79 (±3.32) | 15.77 (±13.84) |
| 10 | BRRI hybrid dhan3 | 32.35 (±1.33) | 39.72 (±1.72) | 22.79 (±1.47) | 21.43 (±2.29) | 30.38 (±1.39) | 41.76 (±19.61) | 57.37 (±1.69) | 58.05 (±0.90) | 1.19 (±4.64) |
| **SEm (±)** | | 1.64 | 1.48 | 9.16 | 1.91 | 2.53 | 18.29 | 2.63 | 2.36 | 8.59 |
| **Mean** | | 24.53 | 26.84 | 9.41 | 20.39 | 24.30 | 19.13 | 51.90 | 53.07 | 1.52 |

treatment in terms of yield parameters, with some varieties exhibiting significant improvements while others show no significant response.

**3.3.3. Reflectance indices.** The impact of the EDU application on alleviating ambient ozone stress varied significantly across different parameters and rice varieties. Among Kasalath-type varieties, significant effects were observed for the parameters simple ratio (SR) and Lichtenthaler indices 2 (Lic2), while other reflectance indices such as normalized difference vegetation index (NDVI), structure intensive pigment index (SIPI), and photochemical reflectance index (PRI) did not show significant responses to the EDU treatment. However, within Kasalath-type varieties, notable responses were observed, indicating a significant interaction between varieties and EDU treatment across all studied reflectance indices parameters. Conversely, in Nipponbare-type varieties, the EDU treatment and its interaction with varieties did not yield significant responses across any of the studied reflectance indices parameters. Nonetheless, significant differences were observed among the varieties themselves for all parameters, suggesting distinct genetic responses irrespective of the EDU treatment (Table 2).

Regarding the parameter simple ratio (SR), indicative of vegetation health, Kasalath-type varieties generally did not exhibit significant responses to the EDU treatment, except for Binadhan-17. Notably, Binadhan-17 displayed a significant response to EDU, possibly influenced by specific alleles or other abiotic factors. Conversely, Nipponbare-type varieties did not show significant responses to the EDU treatment, although there was an overall improvement in leaf greenness,

**(a)**

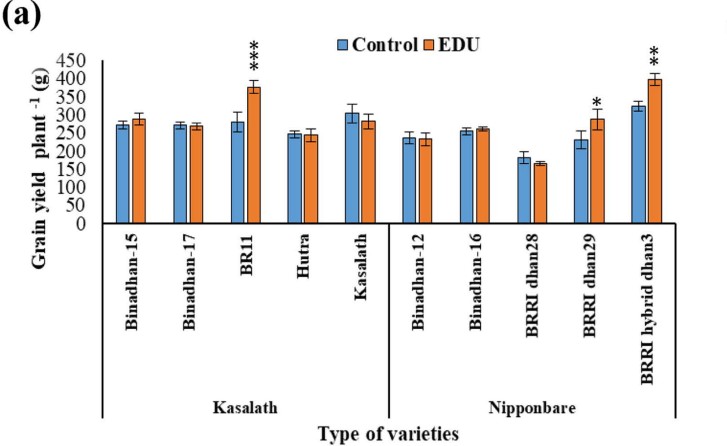

**(b)**

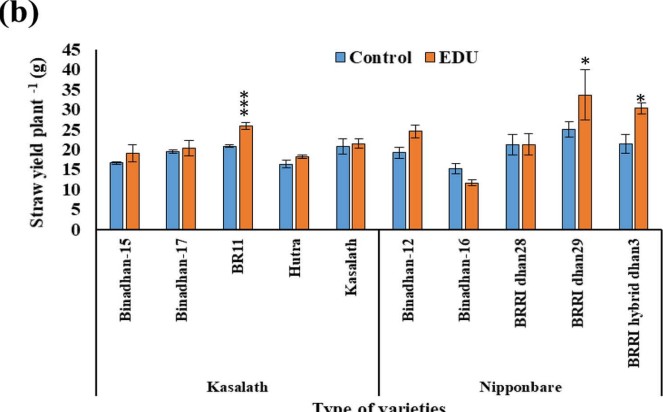

**(c)**

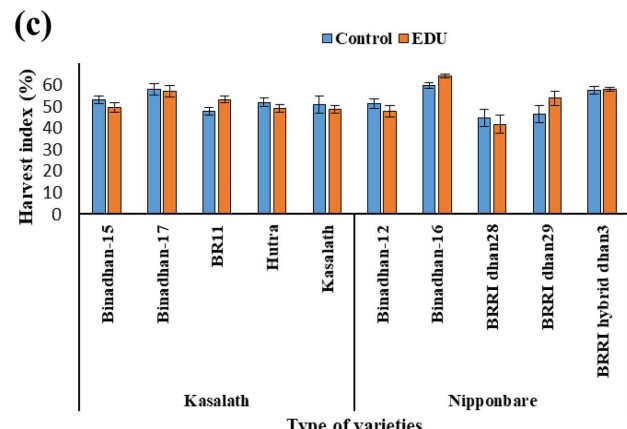

**Fig 4. Yield parameters** [a. Grain yield plant$^{-1}$ b. Straw yield plant$^{-1}$ and c. Harvest index (%)] of 5 Kasalath-type and 5 Nipponbare-type rice varieties grown in ambient ozone concentration with and without application of ethylenediurea (EDU); bars indicate mean values of Mymensingh for irrigated season 2022 with four replicates and standard errors (n = 4); asterisks above the bars indicate significant difference between control and EDU treatment with one variety by Tukey's test (* p < 0.05, **p < 0.01, ***p < 0.001).

except for Binadhan-12, possibly influenced by genetic or environmental factors (Fig 5a). For the structure intensive pigment index (SIPI), estimating the ratio of carotenoids to chlorophyll, both Kasalath-type and Nipponbare-type varieties displayed insignificant responses to the EDU treatment. However, Kasalath-type varieties showed higher responses overall to EDU, except for Kasalath itself. In contrast, Nipponbare-type varieties indicated higher responses to EDU, except for Binadhan-12, suggesting variability in response possibly due to genetic or environmental influences (Fig 5b). "These findings suggest that EDU application can provide some degree of protection against ambient ozone stress in Nipponbare-type varieties. Although the variation for certain reflectance indices under treatment and interaction effects was not statistically significant, trends in leaf greenness and photosynthesis-related parameters indicate a positive influence of EDU. Overall, the study underscores the variability in the response of rice varieties to the EDU treatment in mitigating ambient ozone stress, influenced by genetic factors, environmental conditions, and interactions between varieties and treatments.

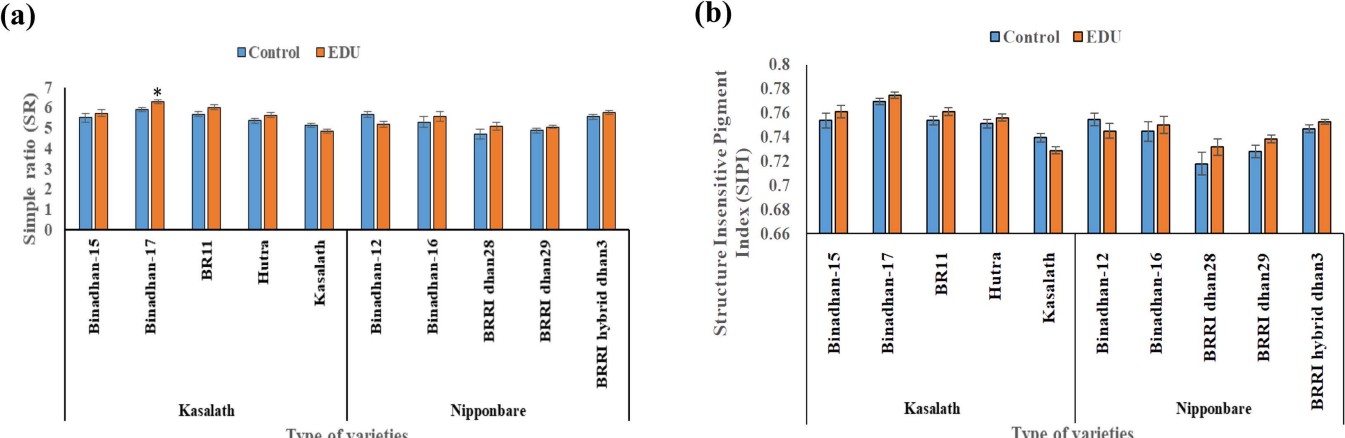

**Fig 5. Physiological traits** [**a.** Simple ratio (SR) and **b.** Structure insensitive pigment index (SIPI)] of 5 Kasalath-type and 5 Nipponbare-type rice varieties grown in ambient ozone concentration with and without application of ethylenediurea (EDU); bars indicate mean values of Mymensingh for irrigated season 2022 with four replicates and standard errors (n = 4); asterisks above the bars indicate significant difference between control and EDU treatment with one variety by Tukey's test (* $p < 0.05$, **$p < 0.01$, ***$p < 0.001$).

## 4. Discussion

### 4.1. Detrimental ambient ozone stress and *OsORAP1* expression in different rice varieties

Rice is an important staple food for more than half of the world's population [41]. However, rice production is threatened by various environmental stresses, such as ozone pollution [42]. Ozone is a major air pollutant that causes significant damage to crops, including rice [43,44]. The rising tropospheric ozone concentrations in South Asia is very explicit [44]. It is one of the most important stress for rice production because rice production is now greatly hampered this region [16]. All of the varieties that we selected for our study were based on Bangladeshi rice production and these are the popular varieties of Bangladesh. To study this, we selected 10 rice varieties where 5 were carrying Kasalath type allele and 5 were carrying Nipponbare type allele. The OzT9 locus, previously identified as a QTL affecting foliar symptom formation in ozone stress, was used as a marker for ozone tolerance [23]. The ozone exposure level is continuously observed in crop-growing environments in South Asia, and daytime ozone peaks exceeding 80 ppb frequently occur, especially from February to April [45]. Still, during the study period, i.e., in irrigated season, it was observed that the daytime ozone peaks in Mymensingh, Bangladesh, exceeded the average 70 ppb (Fig 1). Thus, the ambient ozone exposure in the field is important for breeding ozone-tolerant rice on the South Asian subcontinent, especially Bangladesh. In this study comparing *OsORAP1* expression between Kasalath and Nipponbare rice types under EDU treatment, both types exhibited highly significant differences (P < 0.001) in response to the treatment, as well as in varietal differences and their interactions (Table 1). Further analysis revealed that within the Kasalath-type group, there was generally lower *OsORAP1* expression in both EDU-treated and control varieties, except for the Kasalath variety itself, which showed high tolerance and did not respond to EDU treatment (not responsive to EDU) [39]. On the other hand, the Nipponbare-types displayed higher *OsORAP1* expression in the control varieties compared to EDU-treated ones. Notably, Binadhan-16 and BRRI hybrid dhan3, within the Nipponbare group, exhibited higher *OsORAP1* expression, though these increases were not statistically significant (Fig 2). This suggests a complex interaction between variety type and response to EDU treatment, with Kasalath showing inherent tolerance and Nipponbare-types being more sensitive to the conditions tested.

One of the mechanisms by which plants respond to ozone stress is through the upregulation of the expression of various genes involved in stress tolerance. One such gene is *OsORAP1*, which is involved in oxidative stress response and

has been shown to be upregulated in response to ozone stress in rice. Previous studies have shown that the presence of the QTL affects traits of agronomic importance such as spikelet sterility [46] and chalkiness [47] and the quality of rice straw [48]. Although the QTL has not been fine mapped or cloned to date, experiments with rice mutants suggested that the gene *OsORAP1* may be underlying the QTL effect [7]. We analyzed the *OsORAP1* gene in 20 rice varieties and found that the polymorphisms occurring in the original two contrasting clades are largely conserved. We observed a dramatic difference in terms of foliar symptom formation between the two allele classes, Kasalath type and Nipponbare type (Fig 3). The likely mechanism underlying this phenomenon is that *OsORAP1* is involved in the oxidative cell death cycle, which leads to the induction of programmed cell death in ozone stress [49]. The transcriptional regulation of *OsORAP1* appears to constitute an important switch for the induction of programmed cell death under ozone stress in rice. This information is crucial for rice breeders to develop ozone tolerant rice varieties that can withstand the rising tropospheric ozone concentrations in South Asia.

### 4.2. Response of EDU treatment on different alleles at *OsORAP1* locus for yield and physiological parameters

Tropospheric ozone concentration is increasing throughout the Asian countries due to influence of the activities of human, population growth and rapid development of industry and economy [50]. Thus it is very crucial hazardous gases concisely effecting the agricultural crop production [51] now and even in the future, the crop production will be challenged without using tolerant varieties [22]. Several studies have demonstrated that EDU treatment can effectively mitigate the negative impact of ambient $O_3$ stress on plant growth, yield, and physiological processes of rice [16,31,40]. The underlying mechanism behind the protective role of EDU treatment on plants is thought to be due to its ability to scavenge reactive oxygen species and reduce oxidative damage in plants by protects the surface [52]. The Leaf Bronzing Score (LBS) is commonly employed as a phenotypic measure to assess variances in rice genotypes regarding their tolerance to diverse abiotic stresses, including ozone stress, zinc (Zn) deficiency, salinity, or iron toxicity [31,53,54]. Leaf bronzing score which is the marker of oxidative stress enriched the tolerant mechanism of Kasalath-type *OsORAP1* allele where the Nipponbare-type *OsORAP1* allele boosted the expression of gene in hazardous ozone concentrated area. Similarly, in our study, after application of EDU to the Kasalath-type and Nipponbare-type varieties, Kasalath-type varieties show tolerant, i.e., less/no leaf bronzing score to EDU and insignificant response while the Nipponbare-type showed highly significant response and higher leaf bronzing score (Table 2). So, the Nipponbare-type varieties were EDU responsive, i.e., EDU protects the plants from prolonged cell death [49] and Kasalath-type varieties were less/no EDU responsive, i.e., the plants tolerant mechanism itself protects the plants from visible leaf injury (Fig 3). Different antioxidative defense mechanisms may account for the differential response of these two allele groups to ambient ozone stress. The findings of the present study are consistent with previous research that reported the efficacy of EDU treatment in reducing the adverse effects of ambient $O_3$ stress in plants [55].

EDU treatment in EDU responsive varieties, i.e., Nipponbare type varieties increases the grain yield of rice moderately or not significantly [40,51,55]. In Kasalath-type rice varieties, notable reactions in Kasalath-type *OsORAP1* allele to the application of EDU treatment were noted across various parameters. Conversely, in Nipponbare-type rice varieties, distinct responses to the EDU treatment were observed in Nipponbare-type *OsORAP1* allele, primarily reflected in alterations in the number of unfilled grains plant$^{-1}$ and the resultant grain yield plant$^{-1}$ (Table 2). Focusing specifically on grain yield per plant and straw yield per plant, the response to EDU treatment varied among Kasalath-type rice varieties. Generally, there were no substantial gains or losses observed, with the exception of BR11, which exhibited a notable increase in both parameters following the treatment may be due to genetic or environmental influences, i.e., involvement of some other allele responsible other than *OsORAP1* allele. In contrast, among Nipponbare-type rice varieties, BRRI hybrid dhan3 and BRRI dhan29 showed significant increases in grain yield per plant and straw yield per plant in response to EDU treatment in Nipponbare-type *OsORAP1* allele similar to observations reported by Ashrafuzzaman *et al.* [40] in BRRI dhan28 (BR28), while other varieties did not demonstrate significant changes in either parameter (Table 3, Fig 4a and 4b) and

other varieties did not exhibit statistically significant changes. However, other Nipponbare-type varieties still demonstrated higher grain and straw yields, which may be influenced by genetic or environmental factors. Similarly, the cultivars such as BRRI dhan28 and BRRI dhan29 underwent prior evaluation in controlled environmental trials regarding their reaction to ozone, which were extensively cultivated varieties tailored for the irrigated season, exhibited a yield reduction of 30–40% in open-top chamber experiments when exposed to ozone levels ranging from 77 to 100 ppb throughout the growth period [40,56]. By increasing the number of filled grains per plant, EDU increases the yield of rice varieties [16] with Nipponbare-type *OsORAP1* alleles. EDU treatment has diverse effects on *OsORAP1* alleles in Kasalath and Nipponbare rice varieties, emphasizing the importance of tailoring agricultural practices for optimal yield outcomes. In Table 3, the reduction in yield and related parameters upon EDU treatment in Kasalath-type genotypes (Binadhan-17, Hutra, and Kasalath), as well as in some Nipponbare-type genotypes, may indicate that EDU has little to no protective effect on ozone-tolerant genotypes. These observations underscore the complex interplay of genetic makeup and environmental conditions in determining the physiological responses of rice varieties to agricultural treatments.

The impact of EDU application on mitigating ambient ozone stress varied significantly across parameters and the Kasalath-type and Nipponbare-type rice varieties. To assess the physiological yield response to EDU in both allele varieties, various reflectance indices were gathered. NDVI was employed to assess chlorophyll levels at both the leaf and canopy or ecosystem scales [40]. In our study NDVI, SIPI and PRI were not significant due to EDU treatment in both the allele groups but showed higher response (Table 2) and in previous studies, NDVI was significantly increased by EDU treatment in rice [16], wheat [57] and soybeans [58]. Previously, Lic2 has been proposed as a reliable physiological marker of ozone stress, conducive to high-throughput and non-invasive measurement through spectral reflectance [59]. Similarly, PRI has been recommended as a remote sensing technique for detecting ozone stress in vegetation before visible symptoms manifest [60]. Among Kasalath-type varieties, significant effects were noted for SR and Lic2, with notable interactions observed across all reflectance indices. Conversely, in Nipponbare-type varieties, no significant responses were observed for the studied reflectance indices (Table 2). While SIPI and SR responses were insignificant, Kasalath-type generally exhibited higher responses except Kasalath itself, contrasting with Nipponbare-type (Fig 5a and 5b). These results highlight the nuanced interactions between EDU treatment and rice varieties in combating ozone stress, emphasizing the role of genetic and environmental factors responsible for the EDU response in Kasalath-type and Nipponbare-type varieties. These findings have important implications for the selection of rice varieties for cultivation in areas with high ambient ozone levels, and underscore the need for further research to elucidate the molecular mechanisms underlying the observed differences in response to ozone stress between these two allele groups.

## 5. Conclusions

The response of different rice varieties, specifically Kasalath-type and Nipponbare-type, to Ethylenediurea (EDU) treatment under ambient ozone stress showed distinct genetic responses between these varieties. Nipponbare-type varieties displayed significant increases in *OsORAP1* expression and leaf bronzing score in control plants than the EDU treated plants, while Kasalath-type varieties showed less significant responses due to their inherent tolerance to ozone stress. Additionally, the effects of EDU on yield parameters varied among the varieties. Kasalath-type varieties showed significant responses in certain parameters like the number of ineffective tillers plant[-1] and straw yield plant[-1], whereas Nipponbare-type varieties responded significantly in parameters such as the number of unfilled grains plant[-1] and grain yield plant[-1]. Furthermore, the impact of EDU treatment on alleviating ambient ozone stress differed across reflectance indices parameters, with notable variability observed between Kasalath-type and Nipponbare-type varieties. Overall, the study highlights the intricate interplay of genetic factors, environmental conditions, and treatment interactions in influencing the response of rice varieties to EDU treatment in mitigating ambient ozone stress. The findings underscore the importance of *OsORAP1* in mediating these responses, particularly the contrasting behaviors observed between Kasalath-type and Nipponbare-type varieties. Future research should focus on elucidating the underlying genetic mechanisms, including

the functional role of *OsORAP1*, to better understand and harness these differences. While EDU has proven to be an effective tool in mitigating ozone damage, its limited availability and accessibility pose challenges for widespread agricultural use. Therefore, developing ozone-tolerant rice cultivars through breeding or genetic approaches is essential for ensuring yield stability and the long-term sustainability of rice production in ozone-affected regions.

## Supporting information

**S1 Fig. The DNA profile of 20 rice genotypes using primer KAS_1_2, which showing the band on Kasalath-type (specific) genotypes.** "S1_fig.tif".
(TIF)

**S1 Raw Image Fig. The DNA profile of 20 rice genotypes using primer KAS_1_2, which showing the band on Kasalath-type (specific) genotypes.** "S1_fig.pdf".
(PDF)

**S2 Fig. Photographs of different rice varieties under ambient ozone stress conditions with and without the application of EDU during the irrigated (Boro) season at Mymensingh, Bangladesh (2022). [Shown are images of six rice variety with two treatments: (a) BRRI hybrid dhan3 without EDU (Control), (b) BRRI hybrid dhan3 with EDU, (c) BRRI dhan28 without EDU (Control), (d) BRRI dhan28 with EDU, (e) BRRI dhan29 without EDU (Control), and (f) BRRI dhan29 with EDU. Plants were exposed to ambient ozone stress, and EDU (ethylenediurea) was applied to assess its protective effects against ozone stress].** "S2_fig.tif".
(TIF)

**S1 Dataset. Ambient ozone concentration (ppb)** . "S1 File.xlsx".
(XLSX)

**S2 Dataset. Yield contributing data.** "S2 File.xlsx".
(XLSX)

**S3 Dataset. Spectral reflectance indices data.** "S3 File.xlsx".
(XLSX)

**S4 Dataset. qPCR data for expression analysis.** "S4 File.xlsx".
(XLSX)

## Acknowledgments

The authors thank to Professor Dr. Michael Frei, Department of Agronomy and Crop Physiology, Justus-Liebig-University, Giessen, Germany, for his continuous support including finance for the field experiment. The authors thank to Department of Genetic Engineering & Biotechnology, Shahjalal University of Science and Technology and Bangladesh Institute of Nuclear Agriculture for providing laboratory facilities. The authors would also like to thank German Research Foundation (DFG-Deutsche Forschungsgemeinschaft) for providing a PhD fellowship to Rigyan Gupta.

## Author contributions

**Conceptualization:** Md Ashrafuzzaman.

**Data curation:** Md Ashrafuzzaman.

**Formal analysis:** Mohammad Hasanuzzaman Rani, Md Ashrafuzzaman.

**Investigation:** Rigyan Gupta, Md. Nazmul Hasan.

**Methodology:** Rigyan Gupta, Shamsul H. Prodhan, Md. Nazmul Hasan, Shamsun Nahar Begum, Mirza Mofazzal Islam, Md Ashrafuzzaman.

**Project administration:** Shamsun Nahar Begum, Mirza Mofazzal Islam, Md Ashrafuzzaman.

**Resources:** Shamsul H. Prodhan, Shamsun Nahar Begum, Mirza Mofazzal Islam.

**Software:** Rigyan Gupta.

**Supervision:** Shamsul H. Prodhan, Md. Nazmul Hasan, Shamsun Nahar Begum, Mirza Mofazzal Islam, Md Ashrafuzzaman.

**Validation:** Rigyan Gupta, Shamsul H. Prodhan, Md. Nazmul Hasan, Mirza Mofazzal Islam, Md Ashrafuzzaman.

**Writing – original draft:** Rigyan Gupta, Md. Nazmul Hasan, Mohammad Hasanuzzaman Rani, Md Ashrafuzzaman.

**Writing – review & editing:** Rigyan Gupta, Shamsul H. Prodhan, Md. Nazmul Hasan, Shamsun Nahar Begum, Mohammad Hasanuzzaman Rani, Mirza Mofazzal Islam, Md Ashrafuzzaman.

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
