## [Decision Letter · Decision Letter 0]

Dear Dr. Ashrafuzzaman,

We look forward to receiving your revised manuscript.

Kind regards,

Muhammad Abdul Rehman Rashid, PhD

Academic Editor

PLOS ONE

Journal Requirements:

4. We notice that your supplementary figures are uploaded with the file type 'Other'. Please amend the file type to 'Supporting Information'. Please ensure that each Supporting Information file has a legend listed in the manuscript after the references list.

Reviewers' comments:

Reviewer's Responses to Questions

**Comments to the Author**

1. Is the manuscript technically sound, and do the data support the conclusions?

Reviewer #1: Partly

Reviewer #2: No

2. Has the statistical analysis been performed appropriately and rigorously?

Reviewer #1: N/A

Reviewer #2: Yes

3. Have the authors made all data underlying the findings in their manuscript fully available?

Reviewer #1: Yes

Reviewer #2: No

4. Is the manuscript presented in an intelligible fashion and written in standard English?

Reviewer #1: No

Reviewer #2: No

Reviewer #1: Comment 1: The study included 20 rice varieties, but the criteria for their selection were not clearly defined. For example, whether these varieties are representative of all major cultivar types and whether they contain sufficient genetic diversity are not addressed. Without a comprehensive and representative sample, the generalizability of the study results may be compromised.

Comments 2: Although the experiment included four replications, whether the statistical analysis adequately accounted for random error and replication should be detailed.

Comments 3: EDU is applied weekly by foliar spraying, but specific times and conditions of application (e.g., light, temperature) are not detailed. These factors may greatly influence the effectiveness of EDU.

Comments 4: The experimental procedure is described in detail in the “Materials and Methods” section, but the order in which the results are presented in the “Results” section does not correspond to the experimental design, which may confuse the reader.

Comments 5: Although the study mentions using R software for statistical analysis, the specific methods (e.g., whether multiple comparison corrections were applied) are not detailed. Moreover, for significant results (e.g., p-values), the biological implications are not further explained.

Comments 6: The study does not deeply explore the potential reasons for the differential responses to EDU among varieties. For example, why Kasalath-type varieties show less significant responses to EDU compared to Nipponbare-type varieties is not discussed. Are these differences related to genetic background or environmental adaptability?

Comments 7: The figures in the study could be further optimized. For example, some data labels in the figures are unclear, or sufficient legends are not provided.

Comments 8:When presenting experimental results, some data lack complete statistical information. For example, for certain parameters (e.g., chlorophyll indices), only the mean values are provided without standard deviations or confidence intervals.

Reviewer #2: Comments to the authors

Tropospheric ozone in the atmosphere is considered phytotoxic causing major threat to crop production. Therefore, understanding the effect of O3 in rice plants and its molecular mechanism is essential to manifest tolerance in the rice crop varieties. In this direction, authors have attempted to establish the response of OsORAP1 under O3 stress and EDU treatment.

1. In line 88-90, authors have mentioned “absence of OsORAP1…”. Does this mean OsRAP1 gene does not exist in tolerant genotype.

2. In line 95 – 97, higher induction of OsORAP1 should lead to susceptibility not tolerance.

3. Please provide methodology used for deriving various indices (NDVI, PRI, SIPI etc.)

4. In line 168-170, authors have mentioned “Twenty genotypes (genotypes with Kasalath-type OsRAP1 allele)”. However, all the 20 genotypes were not kasalath-type. This has to be rewritten.

5. In line no 170-173, results of PCR have been described, which may be shifted to results section.

6. Was the primer KAS_1_2 designed in this study or was it sources from literature. If it is designed by the authors, the methodology and the basis has to be elaborated.

7. The KAS_1_2 primer validation should be undertaken between Kasalath and Nipponbare. This is essential step for establishing genotypes as Kasalath-type and Nipponbare-type. However, Nipponbare is missing in the study.

8. Methodology used for scoring leaf bronzing, yield traits and reflectance indices should be included in the materials and methods section

9. In line 118 of materials and methods section, it has been mentioned that 20 genotypes were evaluated in field. However, ANOVA for 10 selected genotypes has been presented.

10. Please shift the content of line no 248-249 to respective discussion section.

11. In fig 3, Kasalath is negatively responding to EDU treatment for leaf bronzing. However, authors have mentioned that Kasalath is not responsive to EDU treatment. This needs to be clarified.

12. In Nipponbare-type varieties, BRRI hybrid dhan3 and BRRI dhan29 exhibited significant increase in grain yield/ plant and straw yield/ plant in response to EDU treatment, while other varieties did not show significant changes in either parameter. This needs to be discussed.

13. Among the Kasalath-type allele carrying varieties, EDU treatment in one of the variety (BR11) has increased the grain and straw yield, while none of the remaining varieties have responded. This needs to be discussed.

14. Association between the reflectance indices with yield traits should be estimated

15. Line No. 327-329, authors have mentioned that “EDU treatment and its interaction with varieties did not yield significant responses across any of the studied reflectance indices parameters. Nonetheless, significant differences were observed among the varieties themselves for all parameters, suggesting distinct genetic responses irrespective of the EDU treatment”. This indicates that there exist inherent differences among the genotypes for the tested traits. This does not imply the variation in genotypes is due to EDU treatment or due to their interaction.

16. In Table 3, among Kasalath-type genotypes, Binadhan 17, Hutra and Kasalath the yield and related parameters have reduced upon EDU treatment over control. Similar situation exists for Nipponbare-type genotypes. This is contradictory. Needs thorough discussion.

17. The OsRAP1 gene expression level is reduced by EDU treatment. However, establishing its association with the yield and other related parameters is essential.

18. The treatment x variety interaction is non-significant for majority of the parameters under study. This indicates that the expression levels of OsRAP1 have no association with any of the tested parameters.

19. In line 335-337, authors have mentioned that “These findings suggest that EDU application can effectively mitigate ambient ozone stress in Nipponbare-type varieties, with different reflectance indices indicating its positive effects on leaf greenness, and photosynthesis content”. The statement is contradicting as the variation for reflectance indices under treatment and interaction component is non-significant.

20. Although, the ANOVA indicates no significant variation for treatment and interaction components for most of the parameters tested. However, authors indicate that there exists influence of treatment and its interaction with genotype although the text. Which is contradictory.

21. Line no 478 and 479 – Authors have mentioned that Nipponbare-type varieties have increased symptom formation and expression under EDU treatment. This contradicts the hypothesis of the paper.

**Do you want your identity to be public for this peer review?** For information about this choice, including consent withdrawal, please see our Privacy Policy

Reviewer #1: No

Reviewer #2: **Yes: ** Ranjith Kumar Ellur

---

## [Author Response · Author response to Decision Letter 1]

23 Mar 2025

Academic editor and Reviewer response

Academic editor

Comments 1: Please ensure that your manuscript meets PLOS ONE's style requirements, including those for file naming. The PLOS ONE style templates can be found at

Author response: Thank you for your valuable comments. We have revised the manuscript to ensure it fully complies with PLOS ONE's style requirements, including formatting and file naming, as per the provided guidelines.

Comments 2: PLOS ONE now requires that authors provide the original uncropped and unadjusted images underlying all blot or gel results reported in a submission’s figures or Supporting Information files. This policy and the journal’s other requirements for blot/gel reporting and figure preparation are described in detail at https://journals.plos.org/plosone/s/figures#loc-blot-and-gel-reporting-requirements and https://journals.plos.org/plosone/s/figures#loc-preparing-figures-from-image-files. When you submit your revised manuscript, please ensure that your figures adhere fully to these guidelines and provide the original underlying images for all blot or gel data reported in your submission. See the following link for instructions on providing the original image data: https://journals.plos.org/plosone/s/figures#loc-original-images-for-blots-and-gels.

Author response: Thank you for your valuable comments. We have revised our submission to fully comply with PLOS ONE’s guidelines. The original uncropped and unadjusted images underlying all blot or gel results have been provided as per the journal’s requirements. In the cover letter we have mention the blot/gel image data are in Supporting Information.

Comments 3: Please include captions for your Supporting Information files at the end of your manuscript, and update any in-text citations to match accordingly. Please see our Supporting Information guidelines for more information: http://journals.plos.org/plosone/s/supporting-information.

Author response: Thank you for your valuable comments. We have revised our submission to fully comply with PLOS ONE’s guidelines. We have included captions for the Supporting Information files at the end of the manuscript and updated all in-text citations accordingly.

Comments 4: We notice that your supplementary figures are uploaded with the file type 'Other'. Please amend the file type to 'Supporting Information'. Please ensure that each Supporting Information file has a legend listed in the manuscript after the references list.

Author response: Thank you for your valuable comments. We have revised our submission to fully comply with PLOS ONE’s guidelines. We have updated the file type of the supplementary figures to 'Supporting Information' and ensured that each Supporting Information file has a corresponding legend listed in the manuscript after the references list.

Reviewer #1

Comments 1: The study included 20 rice varieties, but the criteria for their selection were not clearly defined. For example, whether these varieties are representative of all major cultivar types and whether they contain sufficient genetic diversity are not addressed. Without a comprehensive and representative sample, the generalizability of the study results may be compromised.

Author response: Thank you for your valuable comment. We selected 20 diverse indica rice varieties to ensure genetic variation and broad representation. These included popular mega varieties of Bangladesh where nine inbred varieties developed by the Bangladesh Institute of Nuclear Agriculture (BINA), seven varieties from the Bangladesh Rice Research Institute (BRRI) (including two hybrid and five inbred), three locally cultivated inbred varieties from Bangladesh (including Kasalath, an ozone-tolerant variety), and one variety from the International Rice Research Institute (IRRI). Details are now provided in Table 1 and elaborated in Section 2.1 of the manuscript.

Comments 2: Although the experiment included four replications, whether the statistical analysis adequately accounted for random error and replication should be detailed.

Author response: Thank you for your comment. We used four replications to ensure data reliability. Our statistical analysis accounted for variability, and no significant random errors or differences among replications were observed. Section 2.5 of the manuscript now details our approach to addressing replication in the analysis

Comments 3: EDU is applied weekly by foliar spraying, but specific times and conditions of application (e.g., light, temperature) are not detailed. These factors may greatly influence the effectiveness of EDU.

Author response: Thank you for this insight. We have now included specific details regarding EDU application conditions, such as light and temperature, in Section 2.4 of the manuscript. We have also refined the research objectives for greater coherence and precision.

Comments 4: The experimental procedure is described in detail in the “Materials and Methods” section, but the order in which the results are presented in the “Results” section does not correspond to the experimental design, which may confuse the reader.

Author response: Thank you for your suggestion. We have revised the "Materials and Methods" section for greater clarity and adjusted the "Results" section to enhance logical flow and coherence while maintaining our focus on the relationship between OsORAP1 expression and yield-related data.

Comments 5: Although the study mentions using R software for statistical analysis, the specific methods (e.g., whether multiple comparison corrections were applied) are not detailed. Moreover, for significant results (e.g., p-values), the biological implications are not further explained.

Author response: Thank you for your valuable comment. Section 2.5 now provides details on our statistical methods, including multiple comparison corrections. We have also expanded the discussion on the biological implications of significant results, particularly regarding EDU effects.

Comments 6: The study does not deeply explore the potential reasons for the differential responses to EDU among varieties. For example, why Kasalath-type varieties show less significant responses to EDU compared to Nipponbare-type varieties is not discussed. Are these differences related to genetic background or environmental adaptability?

Author response: Thank you for your insightful comment. We have examined differential responses by analyzing OsORAP1 expression. Previous studies (Ashrafuzzaman et al., 2017, 2018) suggest that tolerant genotypes exhibit minimal response to EDU compared to sensitive ones. This variation is likely due to genetic background, which may influence EDU’s effectiveness in mitigating ozone stress.

Comments 7: The figures in the study could be further optimized. For example, some data labels in the figures are unclear, or sufficient legends are not provided.

Author response: Thank you for your observation. We have revised the figures to improve clarity, including better data labeling and more comprehensive legends for enhanced readability.

Comments 8: When presenting experimental results, some data lack complete statistical information. For example, for certain parameters (e.g., chlorophyll indices), only the mean values are provided without standard deviations or confidence intervals.

Author response: We have reviewed and updated statistical details in Section 2.2, incorporating standard deviations or confidence intervals where necessary. Figure labels have also been improved for better clarity.

Reviewer #2

Comments 1: In line 88-90, authors have mentioned “absence of OsORAP1…”. Does this mean OsRAP1 gene does not exist in tolerant genotype?

Author response: Thank you for your comment. The phrase "absence of OsORAP1" refers to its low or no expression in tolerant genotypes, not the gene’s physical absence. We have revised the text for clarity.

Comments 2: In line 95 – 97, higher induction of OsORAP1 should lead to susceptibility not tolerance.

Author response: Thank you for pointing this out. We have revised the manuscript to accurately convey this concept.

Comments 3: Please provide methodology used for deriving various indices (NDVI, PRI, SIPI etc.)

Author response: Thank you for your valuable comment. We have provided the methodology for deriving various indices (NDVI, PRI, SIPI, etc.) in the "Materials and Methods" section, specifically in Section 2.2 ("Growing of Varieties and Design of the Experiments").

Comments 4: In line 168-170, authors have mentioned “Twenty genotypes (genotypes with Kasalath-type OsRAP1 allele)”. However, all the 20 genotypes were not kasalath-type. This has to be rewritten.

Author response: We apologize for the oversight. The sentence has been revised to: “Among the twenty genotypes, selection was based on the presence or absence of a PCR band for further analyses.”

Comments 5: In line no 170-173, results of PCR have been described, which may be shifted to results section.

Author response: Thank you for your valuable comment. Since our study primarily focuses on the expression of OsORAP1, we included the PCR band results in the "Materials and Methods" section to provide necessary context for our experimental approach. However, we have reconsidered the placement to ensure clarity and coherence in the manuscript.

Comments 6: Was the primer KAS_1_2 designed in this study or was it sources from literature. If it is designed by the authors, the methodology and the basis has to be elaborated.

Author response: The primer KAS_1_2 was sourced from Ashrafuzzaman et al., 2020. This clarification has been added to the manuscript.

Comments 7: The KAS_1_2 primer validation should be undertaken between Kasalath and Nipponbare. This is essential step for establishing genotypes as Kasalath-type and Nipponbare-type. However, Nipponbare is missing in the study.

Author response: Thank you for your suggestion. Prior studies (Ashrafuzzaman et al., 2020) confirm KAS_1_2 amplifies only in Kasalath-type genotypes, while Nipponbare-type primers produce bands exclusively in Nipponbare-type genotypes. We have clarified this in the manuscript.

Comments 8: Methodology used for scoring leaf bronzing, yield traits and reflectance indices should be included in the materials and methods section.

Author response: Thank you for your valuable comment. We appreciate your suggestion and have now included the methodology for scoring leaf bronzing, yield traits, and reflectance indices in the "Materials and Methods" section of the manuscript.

Comments 9: In line 118 of materials and methods section, it has been mentioned that 20 genotypes were evaluated in field. However, ANOVA for 10 selected genotypes has been presented.

Author response: Thank you for your valuable comment. Out of the 20 genotypes evaluated, we selected 10 for further analysis (5 Kasalath-type and 5 Nipponbare-type). Therefore, the ANOVA was conducted based on these 10 selected genotypes, which is why the results are presented accordingly.

Comments 10: Please shift the content of line no 248-249 to respective discussion section.

Author response: Thank you for your valuable comment. We have moved the content from lines 248-249 to the respective discussion section as suggested.

Comments 11: In fig 3, Kasalath is negatively responding to EDU treatment for leaf bronzing. However, authors have mentioned that Kasalath is not responsive to EDU treatment. This needs to be clarified.

Author response: Thank you for your valuable comment. We acknowledge the discrepancy in Figure 3, where Kasalath appears to negatively respond to EDU treatment for leaf bronzing. This response may be due to environmental factors or other variables influencing the outcome. We have clarified this in the manuscript to address the observed effect.

Comments 12: In Nipponbare-type varieties, BRRI hybrid dhan3 and BRRI dhan29 exhibited significant increase in grain yield/ plant and straw yield/ plant in response to EDU treatment, while other varieties did not show significant changes in either parameter. This needs to be discussed.

Author response: Thank you for your valuable comment. In Nipponbare-type varieties, BRRI hybrid dhan3 and BRRI dhan29 showed a significant increase in grain yield per plant and straw yield per plant in response to EDU treatment, while other varieties did not exhibit statistically significant changes. However, other Nipponbare-type varieties still demonstrated higher grain and straw yields, which may be influenced by genetic or environmental factors. We have included this discussion in the manuscript to provide further clarification.

Comments 13: Among the Kasalath-type allele carrying varieties, EDU treatment in one of the variety (BR11) has increased the grain and straw yield, while none of the remaining varieties have responded. This needs to be discussed.

Author response: Thank you for your valuable comment. Among the Kasalath-type allele-carrying varieties, BR11 exhibited an increase in grain and straw yield following EDU treatment, whereas the other varieties did not respond. This variation may be attributed to genetic or environmental factors. We have already addressed this point in the manuscript’s discussion section.

Comments 14: Association between the reflectance indices with yield traits should be estimated.

Author response: Thank you for your valuable comment. We have analyzed the association between reflectance indices and yield traits for both Kasalath-type and Nipponbare-type varieties, with and without EDU treatment, and also have incorporated this information into the manuscript.

Comments 15: Line No. 327-329, authors have mentioned that “EDU treatment and its interaction with varieties did not yield significant responses across any of the studied reflectance indices parameters. Nonetheless, significant differences were observed among the varieties themselves for all parameters, suggesting distinct genetic responses irrespective of the EDU treatment”. This indicates that there exist inherent differences among the genotypes for the tested traits. This does not imply the variation in genotypes is due to EDU treatment or due to their interaction.

Author response: Thank you for your valuable comment. We acknowledge the clarification. Our statement aimed to highlight that while EDU treatment and its interaction with Kasalath-type and Nipponbare-type varieties where it did not show significant effects on reflectance indices, the inherent genetic differences among the varieties were evident. We agree that this variation is independent of EDU treatment.

Comments 16: In Table 3, among Kasalath-type genotypes, Binadhan-17, Hutra and Kasalath the yield and related parameters have reduced upon EDU treatment over control. Similar situation exists for Nipponbare-type genotypes. This is contradictory. Needs thorough discussion.

Author response: Thank you for your valuable comment. In Table 3, the reduction in yield and related parameters upon EDU treatment in Kasalath-type genotypes (Binadhan-17, Hutra, and Kasalath), as well as in some Nipponbare-type genotypes, may indicate that EDU has little to no protective effect on ozone-tolerant genotypes. Additionally, the observed variations could be influenced by environmental conditions or other genetic factors. We further elaborated on this in the discussion section to provide a clearer interpretation.

Comments 17: The OsORAP1 gene expression level is reduced by EDU treatment. However, establishing its association with the yield and other related parameters is essential.

Author response: Thank you for your valuable comment. Throughout our study, we have aimed to establish the association between OsORAP1 gene expression, yield, and other related parameters. We have further clarifi

---

## [Decision Letter · Decision Letter 1]

Dear Dr. Ashrafuzzaman,

Thank you for submitting your manuscript to PLOS ONE. After careful consideration, we feel that it has merit but does not fully meet PLOS ONE’s publication criteria as it currently stands. Therefore, we invite you to submit a revised version of the manuscript that addresses the points raised during the review process.

We look forward to receiving your revised manuscript.

Kind regards,

Muhammad Abdul Rehman Rashid, PhD

Academic Editor

PLOS ONE

Journal Requirements:

Reviewers' comments:

Reviewer's Responses to Questions

**Comments to the Author**

Reviewer #1: All comments have been addressed

Reviewer #3: (No Response)

2. Is the manuscript technically sound, and do the data support the conclusions?

Reviewer #1: Yes

Reviewer #3: Partly

3. Has the statistical analysis been performed appropriately and rigorously?

Reviewer #1: Yes

Reviewer #3: Yes

4. Have the authors made all data underlying the findings in their manuscript fully available?

Reviewer #1: Yes

Reviewer #3: Yes

5. Is the manuscript presented in an intelligible fashion and written in standard English?

Reviewer #1: Yes

Reviewer #3: Yes

Reviewer #1: Since all comments and suggestions have been thoroughly addressed, I recommend acceptance of this manuscript with a few minor revisions, which are optional for the authors.

1. I recommend that the authors consider revising the title of the manuscript to enhance its appeal. A potential title could be “Ethylenediurea inhibits the expression of OsORAP1 in rice, but the protections against ambient ozone vary between varieties”.

2. Page 2, line 19, it is suggested that the authors include a brief introduction about OsORAP1 and EDU to provide context for readers.

3. Page 2, line 27, the phrase “more leaf bronzing score (LBS) following EDU treatment” could potentially mislead readers regarding EDU’s protective effect against ambient ozone. It is recommended to revise it to “more leaf bronzing score (LBS) without EDU protection”.

4. Page 28, line 519, to enhance clarity, it is advisable to replace “EDU treatment” with another expression.

5. For the conclusion, the author should properly highlight the potential impact of their research. For instance, they could mention adapting rice to higher ozone conditions by screening and cultivating tolerant cultivars, and by optimizing the use of EDU and other protectants. This would emphasize the practical implications and significance of their work.

Reviewer #3: The authors need to include suggestions given in attached manuscript PDF.

Abstract: Line 22-23: What level of ppm of EDU was used in this experiment?

Introduction: Add literature review (add 4-5 references from similar previous studies) regarding EDU application in rice.

Provide limitation of this study.

Line 110-111, Keep this highlighted text before aim of this experiment.

Materials and Methods: Provide equation number for formula used.

Results: Table 2: What does these star sign indicate for what information. Write in footnote of this table.

Table 3. Provide SEm along with mean data

Figure 1: Provide name of X-axis. Provide color line graph in Figure 1

Figure 2, Figure 3, Figure 4: Provide color bar graph

References: All references should follow the journal's rules.

**Do you want your identity to be public for this peer review?** For information about this choice, including consent withdrawal, please see our Privacy Policy

Reviewer #1: No

Reviewer #3: **Yes: ** Jiban Shrestha

---

## [Author Response · Author response to Decision Letter 2]

28 May 2025

Academic editor and Reviewer response

Academic editor

Comments 1: Journal Requirements:

Author response: Thank you for your comment regarding the reference list. We have carefully reviewed all cited references in the manuscript to ensure completeness and accuracy.

• We confirm that none of the references cited in the revised manuscript have been retracted.

• Minor corrections have been made to formatting inconsistencies and citation details (e.g., author names, journal names, volume/page numbers) to ensure compliance with journal guidelines.

• A few outdated references have been replaced with more recent and relevant literature to strengthen the manuscript's background and discussion sections.

All changes to the reference list have been clearly marked in the revised manuscript, and these updates are detailed in the point-by-point responses where applicable.

Thank you for your guidance.

Reviewers' comments:

Reviewer #1

Comment 1: I recommend that the authors consider revising the title of the manuscript to enhance its appeal. A potential title could be “Ethylenediurea inhibits the expression of OsORAP1 in rice, but the protections against ambient ozone vary between varieties”

Author response: We appreciate the reviewer’s thoughtful suggestion regarding the manuscript title. After careful consideration, we have revised the title to better reflect the key findings and improve the manuscript's appeal. Please see below the new title which we amended in the manuscript:

“Ethylenediurea (EDU) inhibits OsORAP1 expression in rice (Oryza sativa L.): varietal differences in ozone protection efficacy”

This revised title captures both the molecular and varietal aspects of the study and aligns well with the reviewer's recommendation.

Comments 2: Page 2, line 19, it is suggested that the authors include a brief introduction about OsORAP1 and EDU to provide context for readers.

Author response: We thank the reviewer for this valuable suggestion. In response, we have revised the sentence on Page 2, line 19 to include a brief introduction to both OsORAP1 and EDU. The revised sentence now reads:

“This study investigates the response of the OsORAP1 allele—an important regulator of the plant's response to ozone stress, whose expression is associated with ozone-induced damage—in different rice varieties, particularly Kasalath-type and Nipponbare-type, under treatment with ethylenediurea (EDU), a protective antiozonant used in plant research under ambient ozone stress.”

This revision provides the necessary background to help readers better understand the context of our study.

Comments 3: Page 2, line 27, the phrase “more leaf bronzing score (LBS) following EDU treatment” could potentially mislead readers regarding EDU’s protective effect against ambient ozone. It is recommended to revise it to “more leaf bronzing score (LBS) without EDU protection”.

Author response: We thank the reviewer for pointing out the potential for misinterpretation. To accurately reflect EDU's protective role, we have revised the phrase on Page 2, line 27 to read:

“more leaf bronzing score (LBS) without EDU protection.”

This correction ensures clarity and better aligns with the intended meaning regarding the effects of ozone stress and the role of EDU.

Comments 4: Page 28, line 519, to enhance clarity, it is advisable to replace “EDU treatment” with another expression.

Author response: Thank you for your helpful comment. We have revised the sentence to enhance clarity. It now reads:

“Nipponbare-type varieties displayed significant increases in OsORAP1 expression and leaf bronzing score in control plants than the EDU-treated plants, while Kasalath-type varieties showed less significant responses due to their inherent tolerance to ozone stress.”

This revision clarifies the comparison between control and EDU-treated plants and aligns better with the intended meaning.

Comments 5: For the conclusion, the author should properly highlight the potential impact of their research. For instance, they could mention adapting rice to higher ozone conditions by screening and cultivating tolerant cultivars, and by optimizing the use of EDU and other protectants. This would emphasize the practical implications and significance of their work.

Author response: We thank the reviewer for this insightful suggestion. In response, we have revised the conclusion to better highlight the practical implications of our findings. Specifically, we now emphasize the potential for adapting rice to increasing ambient ozone levels through the development and cultivation of ozone-tolerant varieties. We also discuss the role of EDU and other protectants as mitigation tools for ozone stress in sensitive varieties, while noting their limitations in terms of availability and use in agricultural practice. These additions underscore the broader significance of our research for crop improvement and the sustainability of rice production under changing environmental conditions.

Reviewer #3

Comments: Abstract: Line 22-23: What level of ppm of EDU was used in this experiment?

Author response: Thank you for pointing this out. We have now clarified the concentration of EDU used in the experiment in the Abstract section. Specifically, we used 300 ppm of EDU, applied as a foliar spray at regular intervals. The revised sentence now reads:

"The experiment, conducted during the 2022 rainfed season, involved growing of 20 rice varieties under a split-plot design with and without 300 ppm EDU treatment, followed by genomic DNA collection at the vegetative stage to differentiate Kasalath-types and Nipponbare-types, and RNA extraction from 10 selected varieties at the flowering stage for gene expression analysis."

This correction has been incorporated into the revised manuscript.

Comments: Introduction: Add literature review (add 4-5 references from similar previous studies) regarding EDU application in rice. Provide limitation of this study.

Author response: We thank the reviewer for this valuable suggestion. In the revised manuscript, we have expanded the Introduction section to include a brief literature review on the use of ethylenediurea (EDU) in rice studies. We have incorporated references to relevant previous research that demonstrates the protective role of EDU against ozone-induced stress in rice. The revised section now includes the following studies:

Previous studies have explored the use of ethylenediurea (EDU) as a protective agent against ozone stress in rice. Several researchers have reported that EDU can enhance rice yield, primarily through improvements in grain weight rather than increases in the number of panicles or grains per plant (Shang et al., 2022). Under conditions of elevated surface ozone, EDU has been shown to provide moderate mitigation of yield loss, suggesting its potential as a practical tool for safeguarding rice productivity (Zhang et al., 2022; Ashrafuzzaman et al., 2018). Interestingly, its effectiveness appears to vary between rice genotypes, with some evidence indicating that inbred cultivars benefit more from EDU treatment than hybrid varieties (Zhang et al., 2010). In addition to its physiological effects, EDU application does not appear to significantly alter gene expression patterns, highlighting its suitability for field-based ozone screening without interfering with molecular responses. More recently, Frei et al. (2024) demonstrated that EDU can serve as a diagnostic tool to differentiate between ozone-sensitive and ozone-tolerant rice genotypes, with yield increases observed up to 21% in responsive varieties. These findings collectively underscore the potential of EDU in both protecting rice from ozone stress and facilitating the identification of tolerant cultivars. These studies provide a strong background for our work and highlight the rationale for using EDU in our experiment.

In addition, we have now clearly stated the limitation of the present study. While the study identifies ozone-tolerant rice varieties and explores their genetic expression profiles, it is limited to a single growing season and location. Furthermore, the effect of EDU was studied at only one concentration (300 ppm), and the long-term agronomic and physiological impacts under varying field conditions were not assessed. These aspects should be explored in future multi-location and multi-season trials.

Comments: Line 110-111, Keep this highlighted text before aim of this experiment.

Author response: Thank you for your suggestion. As per your recommendation, the sentence "The study was conducted based on 10 Bangladeshi rice varieties during their flowering stage with and without the application of EDU (300 ppm, once a week after transplanting until harvesting)" has been moved to appear before the statement outlining the aim of the experiment. This revision improves the logical flow of the methodology and clarifies the experimental context prior to presenting the research objective.

Comments: Materials and Methods: Provide equation number for formula used.

Author response: Thank you for your valuable comment. We have added equation numbers to all the formulas used in the manuscript to enhance clarity and ease of reference.

Comments: Results: Table 2: What does these star sign indicate for what information. Write in footnote of this table.

Author response: Thank you for pointing this out. We have now clarified the meaning of the star signs in Table 2. Specifically, the asterisks indicate the level of statistical significance, as follows:

*p < 0.05, **p < 0.01, ***p < 0.001.

We have added this information as a footnote to Table 2 in the revised manuscript to ensure clarity for readers.

Comments: Table 3. Provide SEm along with mean data

Author response: Thank you for your valuable suggestion. We have now included the Standard Error of the Mean (SEm) alongside the mean values in Table 3 to provide a better understanding of the data variability. The revised table with SEm values is now incorporated in the updated manuscript.

Comments: Figure 1: Provide name of X-axis. Provide color line graph in Figure 1

Author response: Thank you for your helpful suggestions. We have revised Figure 1 to address both points:

X-axis Label: We have added an appropriate label to the X-axis to clearly indicate the variable being represented.

Color Line Graph: We have converted the graph to a color format to enhance visual clarity and distinguishability of the data series.

The updated figure has been included in the revised manuscript (see revised Figure 1). We hope these improvements meet your expectations.

Comments: Figure 2, Figure 3, Figure 4: Provide color bar graph

Author response: Thank you for your valuable feedback. We have revised Figures 2, 3, and 4 to present the bar graphs in color. This modification enhances the visual clarity and makes it easier to differentiate between data groups. The updated figures have been incorporated into the revised manuscript.

We appreciate your suggestion, which has helped improve the overall presentation of our data.

Comments: References: All references should follow the journal's rules.

Author response: Thank you for pointing this out. We have carefully reviewed and revised all references to ensure they fully comply with the journal’s formatting guidelines. This includes adjustments to author names, journal titles, volume and issue numbers, page ranges, and DOI formatting where applicable. We appreciate your attention to detail in helping us improve the manuscript’s consistency.

---

## [Editor Report · Decision Letter 2]

Ethylenediurea (EDU) inhibits OsORAP1 expression in rice (Oryza sativa L.): varietal differences in ozone protection efficacy

PONE-D-25-00588R2

Dear Dr. Ashrafuzzaman,

We’re pleased to inform you that your manuscript has been judged scientifically suitable for publication and will be formally accepted for publication once it meets all outstanding technical requirements.

Kind regards,

Muhammad Abdul Rehman Rashid, PhD

Academic Editor

PLOS ONE

---

## [Editor Report · Acceptance letter]

PONE-D-25-00588R2

PLOS ONE

Dear Dr. Ashrafuzzaman,

I'm pleased to inform you that your manuscript has been deemed suitable for publication in PLOS ONE. Congratulations! Your manuscript is now being handed over to our production team.

Kind regards,

on behalf of

Dr. Muhammad Abdul Rehman Rashid

Academic Editor

PLOS ONE